

# Exposure to low concentrations of pesticide stimulates ecological functioning in the dung beetle *Onthophagus nuchicornis*

Paul Manning and G. Christopher Cutler

Department of Plant, Food, and Environmental Sciences, Dalhousie University, Truro, Nova Scotia, Canada

## ABSTRACT

Body-size is an important trait for predicting how species contribute to ecosystem functions and respond to environmental stress. Using the dung beetle *Onthophagus nuchicornis* (Coleoptera: Scarabaeidae), we explored how variation in body-size affected ecosystem functioning (dung burial) and sensitivity to an environmental stressor (exposure to the veterinary anthelmintic ivermectin). We found that large beetles buried nearly 1.5-fold more dung than small beetles, but that mortality from exposure to a range of concentrations of ivermectin did not differ between large and small beetles. Unexpectedly, we found that exposure to low concentrations of ivermectin (0.01–1 mg ivermectin per kg dung) stimulated dung burial in both small and large beetles. Our results provide evidence of ecological functioning hormesis stemming from exposure to low amounts of a chemical stressor that causes mortality at high doses.

## INTRODUCTION

Insect species vary in their sensitivity and responses to environmental disturbances. The framework of "response traits" is useful for understanding mechanisms behind this varied sensitivity (*Hevia et al., 2017*). Response traits are phenotypic or life history traits that can determine species sensitivity to disturbance. For example, bee species that nest below-ground are more susceptible to soil tillage than species that nest above ground (*Williams et al., 2010*), and carnivorous carabid beetles are more susceptible to landscape simplification than phytophagous or omnivorous species (*Purtauf, Dauber & Wolters, 2005*)

Loss of insect diversity and abundance caused by environmental disturbances affects the functioning of natural and managed ecosystems (*Larsen, Williams & Kremen, 2005*; *Winfree et al., 2015*). A useful framework to predict and understand the implications of declining abundance and diversity on ecosystem functioning is that of "effect traits"; traits that influence a species contribution to an ecosystem function (*Lavorel & Garnier, 2002*). For example, the ability of a bee to successfully pollinate crops with accessible

Corresponding author
Paul Manning, paul.manning@dal.ca

flowers increases as tongue length decreases (*Williams et al., 2010*), and the ability of carabid beetles to efficiently kill and consume prey increases with body size (*Ball et al., 2015*).

Response traits and effect traits are sometimes interlinked where the traits that make a species susceptible to an environmental disturbance also make that species more or less important in providing an ecosystem function (*Nervo et al., 2017*). As the field of trait-based research continues to grow, this relationship has become more intensively studied (*Hevia et al., 2017*). Though most research on the interplay of effect and response traits explores change at the ecosystem or community level, considerable trait variability also occurs within species. Intraspecific trait variation is generated through a combination of genetic and environmental factors and includes body-size, stress tolerance, and personality. Variation in these traits has long been understood to have implications for an individual's fitness, though variation also has significant bearing for ecosystem functioning (*Bolnick et al., 2011*).

Here, we considered the importance of intraspecific trait variation by exploring the effect of body-size as both an effect trait and response trait in the dung beetle *Onthophagus nuchicornis* (Coleoptera; Scarabaeidae). *Onthophagus nuchicornis* is a generalist coprophage native to the Palearctic that has become widely naturalized within North America (*Floate & Kadiri, 2013*). In northern North America, it is often highly abundant, representing 80% or more of the dung beetle community at latitudes above 44N in Western USA (*Jones et al., 2019*), and approximately 40% of the dung beetles in native grasslands in Alberta (*Floate & Kadiri, 2013*). As a tunneling species, *O. nuchicornis* buries dung in soil and fashions it into oblong parcels known as "brood balls". Females lay a single egg within each brood ball where after hatching a larva will feed for the entirety of its development (*Skidmore, 1991*). Burial and subsequent decomposition of dung by *Onthophagus* spp. supports ecosystem functions including reductions in abundance of blood-feeding flies (*Macqueen & Beirne, 1975b*), improved soil fertility (*Macqueen & Beirne, 1975a*), stimulation of plant litter decomposition (*Manning et al., 2016*), and secondary seed dispersal (*Koike et al., 2012*).

Within agroecosystems, dung beetles and the ecosystem functions they support are vulnerable to veterinary parasiticides. When parasiticides, including macrocyclic lactones and synthetic pyrethroids, are applied to livestock they are excreted in dung largely unmetabolized (*Lumaret et al., 2012*). The compound which has perhaps received the most attention is ivermectin. Ivermectin exposure can increase nerve and muscle cell permeability to chloride ions, leading to hyperpolarization that limits action potentials (*Shoop & Soll, 2002*), and subsequently loss of olfaction, reduced fecundity, impaired movement, and death of dung beetles (*Verdú et al., 2015*; *Martínez et al., 2017*).

While studies have explored the potential of body-size on sensitivity to environmental disturbances across entire dung beetle communities (*Tonelli, Verdú & Zunino, 2018*), to the best of our knowledge none have considered how intraspecific trait variation influences sensitivity of dung beetles to environmental disturbances. In laboratory

experiments using the dung beetle *O. nuchicornis*, we examined the importance of intraspecific body size as a response trait (sensitivity to the veterinary anthelmintic ivermectin) and an effect trait (burial of sheep dung).

We predicted that:

a) Intraspecific variation in body size is a relevant effect trait, such that larger beetles would bury more dung than smaller beetles.

b) Intraspecific variation in body size is not a relevant response trait. Because beetles are exposed to ivermectin through contact and oral exposure as they feed and bury within dung, exposure occurs independently of body size. Larger beetles will be similarly susceptible to ivermectin exposure as smaller beetles.

c) Because low level exposure to ivermectin is known to strongly and negatively influence beetle condition, the importance of body size as an effect trait will diminish when beetles are exposed to ivermectin.

## MATERIALS AND METHODS

### Collection and maintenance of beetles

With permission from landowners, we collected *O. nuchicornis* from two equine facilities in the vicinity of Truro, Nova Scotia (Camden Stables, Opportunity Farm) between 24 and 27 June 2019. The horses had not been treated with any sort of parasiticides within the previous four months at either of the two locations. *Onthophagus nuchicornis* is univoltine and undergoes an obligate reproductive diapause while overwintering as an adult (*Floate et al., 2015*). Because May and June were especially cold, all beetles we collected were assumed to be overwintered sexually mature adults.

Upon return to the laboratory, beetles were housed intermittently in vented 11.4 L polypropylene containers. To each container, we added two dozen crumpled sheets of unbleached paper towel that were wetted with distilled water and 250 g of horse dung from a facility (Camden Stables) where we collected beetles. An additional 100 g horse dung was added every three days until beginning the experiment.

### Dung collection and preparation

Fresh sheep dung used in the experiment was collected from a herd of ewes housed indoors at the Dalhousie University Agricultural Campus. Sheep were fed a diet of hay supplemented with a 80:20 barley:high protein pellet. None of the sheep had been treated with any parasiticides within eight months. Fresh sheep dung was collected 14–24 June 2019. Each day following collection the dung was homogenized for 10 minutes using a paint mixer (Mastercraft, Toronto, Canada) attached to a cordless drill and divided into two equal portions. Dung was frozen at −21 °C prior to use in the experiment. Twenty-four hours prior to beginning the experiment, half the dung collected was removed from the freezer and thawed at room temperature. After the dung was thoroughly homogenized it was weighed and hand-rolled into 100 g (+/− 2 g) balls.

 

## Enclosure design

Experimental enclosures (herein containers) were assembled using 1,360 mL grease resistant paper cups (Dart Container Corporation, Mason, MI, US). Eight 2 mm drainage holes were made in the bottom of each container. Each container was filled with coarse all-purpose builder's sand (Shaw Resources, Shubenacadie, Canada) to a depth of 12 cm leaving 5.8 cm between the sand surface and the mouth of the container. Distilled water was slowly added to each container until water began to drip from the drainage holes. Dung was placed on top of the sand and pressed gently against the sand surface to ensure contact.

## Sex and size classification of beetles

We collected 358 beetles (170 males, 188 females) from the field. Males were differentiated from females by the presence of a single spine-like horn on the head; females had a transverse ridge at the base of the head and no horn (*Jessop, 1986*). Cohorts of females and males were placed in separate holding containers partially filled with moist paper towel.

We labeled 90 mL plastic specimen cups filled half-way with moist all-purpose builder's sand ($N = 60$). Cups were labeled from S1 to S30 and L1 to L30, coding for small and large beetles, respectively. We then created our sub-population of male beetles by selecting the largest individuals based on sight and placing them individually in cups L1–L30. The order of the cups was randomized, and a second large male beetle was added to each cup. We repeated the same process for large female beetles, such that each cup contained four beetles (two males and two females) sampled from the larger end of the size distribution. Immediately after the last beetle was added, the container was closed with 2-mm black fiberglass mesh secured by an elastic band. Immediately after, the same process was followed for small beetles placing them in cups labeled from S1 to S30. All remaining beetles were released.

## Estimating the importance of body size in functioning

We added beetles from cups into containers on 28 June. Beetles were left to bury and feed on dung for seven days. On 5 July we gently sifted the contents of each container through a 0.4 cm aperture soil sieve. After sifting, all beetles were gently removed and placed in a 90 mL specimen cup, and all brood balls were placed in a Petri dish. We recorded the number of brood balls formed in each container and measured the cumulative mass of buried dung to the nearest tenth of a gram. We measured the mass of each beetle to the nearest milligram. Three beetles died in the first part of the experiment and were replaced with beetles of the same size and sex housed under identical conditions.

## Estimating intraspecific variation in functioning under chemical perturbation

The same beetles were used for the second experiment, which began immediately after the first experiment. Dung was defrosted 48 h prior and was subsequently homogenized for 10 min as described above. We subdivided the dung into six 1.0 kg portions. We added 100 mL of Ivomec, Pour-On for Cattle (5 mg·mL$^{-1}$ ivermectin) diluted in acetone to

each portion of dung. The final treatment concentrations of ivermectin in dung were 0.01, 0.1, 1, 10 and 100 mg·kg$^{-1}$ (wet weight). A 100 mL solution of pure acetone was added to the remaining 1.00 kg portion of dung to serve as a control treatment. As before, we homogenized dung using an electric paint mixer and divided each parcel into $n = 10$ dung balls each measuring 100 g. To avoid contamination, when mixing ivermectin into the dung we moved sequentially from ivermectin-free controls to the highest concentration of ivermectin. The paint mixer attachment was thoroughly cleaned in hot soapy water, rinsed, and dried between mixing different batches of dung. We left all dung in a well-ventilated area for 24 h to allow acetone to volatize.

We began the second experiment as the first experiment (described in "Estimating the importance of body size in functioning") was ending (5 July). As beetles and their brood were removed from each container, we repacked sand into each enclosure and re-wetted as before. Dung balls containing ivermectin were split amongst size classes (e.g., $n = 5$ "large beetle" containers with 10 mg·kg$^{-1}$ ivermectin and $n = 5$ "small beetle" containers with 10 mg·kg$^{-1}$ ivermectin) that were pre-assigned using a random number generator. Levels of ivermectin exposure were then randomly assigned to containers. We added a dung ball to each container, pressing the dung gently against the soil surface to ensure contact between sand and dung. Beetles were added to containers immediately after being weighed and each container was closed with 2 mm black fiberglass mesh and secured with two rubber bands to ensure beetles could not escape.

Following a seven-day period (12 July) the experiment was completed. We were blinded from the concentration of the ivermectin during observation. As before, we used a sieve to separate beetles and brood balls from sand. We recorded the number and cumulative mass of brood balls within each enclosure. We recorded the sex of each beetle, and whether it was dead or alive. A beetle was considered dead if gentle prodding with a pair of blunt forceps, or if breathing onto the beetle failed to elicit any significant antennal or leg movements. Beetles were subsequently euthanized by freezing at −21 °C, and later a second estimate of body size (pronotal width) was taken using digital calipers to the nearest hundredth of a millimeter. While measuring pronotal width, we detected an error in differentiating male and female beetles that occurred in three replicates. The final sample size was reduced from $N = 60$ to $N = 57$.

## Statistical analysis

To confirm that body size differed between small and large groups, we used a two-sample $t$-test that compared pronotal width and mass between the two size classifications. Because *O. nuchicornis* is sexually dimorphic, we completed separate tests for males and for females.

To determine whether intraspecific variation in body size is a relevant effect trait, we compared small and large beetles across three measures of functioning. We used a two-sample t-test to compare mean dung ball mass and cumulative mass of dung buried. We used a linear model with a Poisson error structure to determine if the number of brood balls varied between small and large beetles.

To determine if intraspecific variation in body size is a relevant response trait, we initially intended to compare by probit analysis the sensitivity of beetles between the size classes by calculating the ivermectin concentration that kills 50% of the population. This was not possible because of relatively high (15 ± 5%) mortality in the control group and failure to select an experimental dose high enough to generate the full sigmoidal dose-response curve. We instead treated dose as a categorical factor and compared mortality across the full range of doses using generalized linear models with a binomial error structure. Where overdispersion occurred, models were compared using generalized linear model with quasi-binomial error structure. Statistical significance was calculated by comparison to a null, intercept-only, model using an "F" test for models with binomial error, and a "Chi-Square" test for models with quasibinomial error.

Lastly, to test whether ecosystem functioning provided by beetles with a larger body size would be less impaired by ivermectin in comparison to beetles with a smaller body size, we used generalized linear models. Ivermectin concentration (categorical: $0.01–100$ mg·kg$^{-1}$) and body size (categorical: small vs. large) were used as independent variables along with their interaction. As above, we used three different estimates of functioning as dependent variables: dung removal (Gaussian error structure), mean brood ball size (Gaussian error structure), and brood ball number (Poisson error structure).

All statistical models were run using R 3.6.2 (*R Core Team, 2016*), along with the packages, dplyr (*Wickham et al., 2015*) and ggplot2 (*Wickham, 2016*). When reporting results, in all cases we have used means and associated standard errors.

## RESULTS

### Beetles varied significantly in size between population samples

Our efforts to select a sample of large and small beetles from the larger population were effective. The mean mass of large males (61.8 ± 1.0 mg) was 46% heavier than the mass of small males (42.2 ± 0.8 mg, $t_{112} = 16.4$, $P < 0.001$, Fig. 1A). The pronotal width of large males (4.51 mm ± 0.02 mm) was 15% greater than the pronotal width of small males (3.93 ± 0.02 mm, $t_{112} = 18.8$, $P < 0.001$, Fig. 1B). Mean mass of large females (64.6 ± 1.1 mg) was 84% greater than the mass of small females (34.3 ± 0.8 mg, $t_{112} = 21.5$, $P < 0.001$, Fig. 1C). The pronotal width of large females (4.53 ± 0.02 mm) was nearly 25% greater than the pronotal width of small females (3.61 ± 0.03 mm, $t_{112} = 22.8$, $P < 0.001$, Fig. 1D).

### Intraspecific body size is a highly relevant effect trait

Larger beetles buried 1.5-fold more dung (47.6 ± 2.8 g) than smaller beetles (19.0 ± 2.3 g, $t_{54} = 7.84$, $P < 0.001$, Fig. 2A). Larger beetles buried nearly twice as many brood balls (17.1 ± 0.9 g) in comparison to smaller beetles (9.6 ± 1.1, $F_{1,55} = 20.33$, $P < 0.001$, Fig. 2B).

The mean mass of brood balls formed by large beetles (2.8 ± 0.1 g) was 38% heavier than the mean mass of brood balls formed by smaller beetles (2.0 ± 0.1 g, $t_{51} = 5.80$, $P < 0.001$, Fig. 2C).
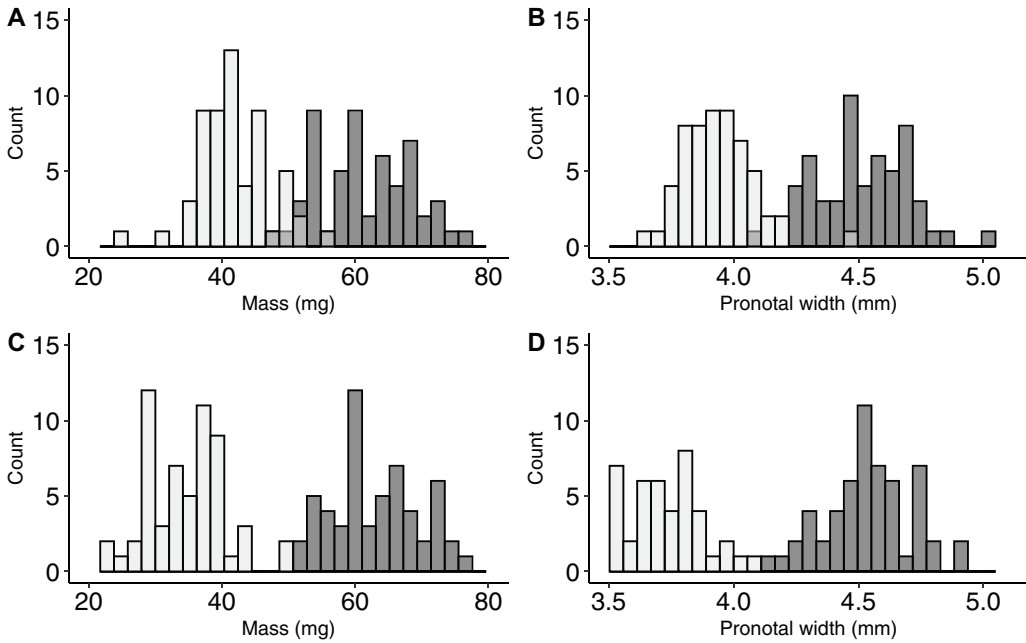

**Figure 1 Demonstrating the large differences in body-size for *Onthophagus nuchicornis*.** Distributions in body-size variation of *Onthophagus nuchicornis* across male body mass (A), male pronotal width (B), female body mass (C), and female pronotal width (D). Beetles from the smaller end of the body-size spectrum are filled in light gray. Beetles from the larger end of the body-size spectrum are filled in dark gray. In all cases differences in means were statistically significant with large effect sizes (minimum 15% difference).

## Beetles susceptibility to ivermectin does not vary with body size

We found no significant interaction between body-size and exposure level on beetle mortality ($\chi_{5,45} = 3.80$, $P = 0.58$). While beetle sensitivity to ivermectin was not significantly affected by body size ($\chi_{1,54} = 0.51$, $P = 0.48$), the level of ivermectin exposure significantly affected survival ($\chi_{5,50} = 54.46$, $P < 0.001$, Fig. 3). Beetle survival in the control group was 85 ± 6%, increasing to 97 ± 3%, 93 ± 4% and 97 ± 3% at 0.01, 0.1 and 1 mg ivermectin·kg$^{-1}$ dung respectively. Survival dropped to 68 ± 13% at 10 mg ivermectin·kg$^{-1}$ dung and 8 ± 4% at 100 mg ivermectin·kg$^{-1}$ dung.

## Exposure to low concentrations of ivermectin stimulates functioning

Total dung buried was significantly affected by beetle size ($F_{1,54} = 4.16$, $P = 0.048$) and by ivermectin exposure ($F_{5,49} = 3.00$, $P = 0.004$), but not by the interaction of these factors ($F_{5,44} = 1.00$, $P = 0.43$) (Fig. 4A). Low level exposure to ivermectin stimulated functional efficiency. Relative to dung buried by beetles not exposed to ivermectin (1.16 ± 0.57 g), beetles exposed to 0.01 mg·kg$^{-1}$ ivermectin buried 13.7-fold more dung (15.9 ± 8.4 g), beetles exposed to 0.1 mg·kg$^{-1}$ ivermectin buried 10.8-fold more dung (12.6 ± 3.6 g), and beetles exposed to 1 mg·kg$^{-1}$ ivermectin buried 6-fold more dung (6.9 ± 1.9 g). At higher concentrations dung burial was negatively affected where exposure to 10 mg·kg$^{-1}$ ivermectin reduced dung burial by 54% (0.53 ± 0.4 g) and exposure to 100 mg·kg$^{-1}$ ivermectin reduced dung burial by 81% (0.22 ± 0.2 g). Across all levels of

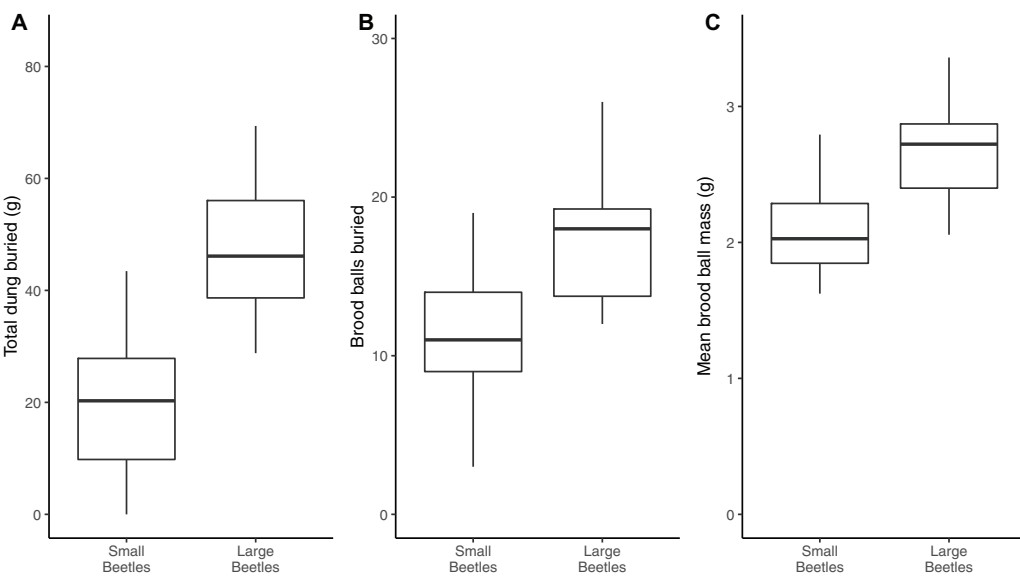

**Figure 2 Larger dung beetles are more functionally efficient than smaller dung beetles.** Comparison of three different measures of functional efficiency in response to body size in the dung beetle *Onthophagus nuchicornis*. Body size is an important predictor of functional efficiency. Larger beetles bury more dung than small beetles (A), which is facilitated by larger beetles burying a larger number of brood balls than smaller beetles (B) in greater abundances (C). The lower and upper limits of the box represent the inter quartile range (IQR) of the data. The horizontal line within each box represents the median. Whiskers extend from the boxes to show maximum and minimum values.

ivermectin exposure, larger beetles buried 8.6 ± 3.0 g of dung, which was approximately 1.75-fold more dung than smaller beetles (3.1 ± 0.9 g).

We found a similar trend for brood ball production (Fig. 4B). The number of brood balls buried per mesocosm was significantly affected by beetle size ($F_{1,55} = 14.36$, $P = 0.01$) and by ivermectin exposure ($F_{5,50} = 12.20$, $P < 0.001$), but not by their interaction ($F_{5,45} = 4.31$, $P = 0.82$). Low level exposure to ivermectin stimulated production of brood balls. Relative to the number of brood balls produced by beetles that were not exposed to ivermectin (0.6 ± 0.3 brood balls), beetles exposed to 0.01 mg·kg$^{-1}$ ivermectin buried 9-fold (5.4 ± 2.4 brood balls) more brood balls, and beetles exposed to 0.1 mg·kg$^{-1}$ ivermectin buried more 8-fold more (4.8 ± 1.2 brood balls, Fig. 1A). Across all levels of ivermectin exposure, larger beetles buried twice as many brood balls (3.0 ± 0.9) as smaller beetles (1.4 ± 0.4 brood balls).

The mean size of brood balls (Fig. 4C) was not affected by beetle size ($F_{1,27} = 2.82$, $P = 0.11$,), ivermectin exposure ($F_{5,22} = 0.33$, $P = 0.89$), or the interaction of these factors ($F_{3,19} = 0.50$, $P = 0.69$).

## DISCUSSION

Beetles with a larger body-size were more efficient at removing dung, suggesting body-size is an important intraspecific effect trait for *O. nuchicornis*. This trend has been observed for other species of dung beetles including *Onthophagus* (*Lee & Peng, 1981*).

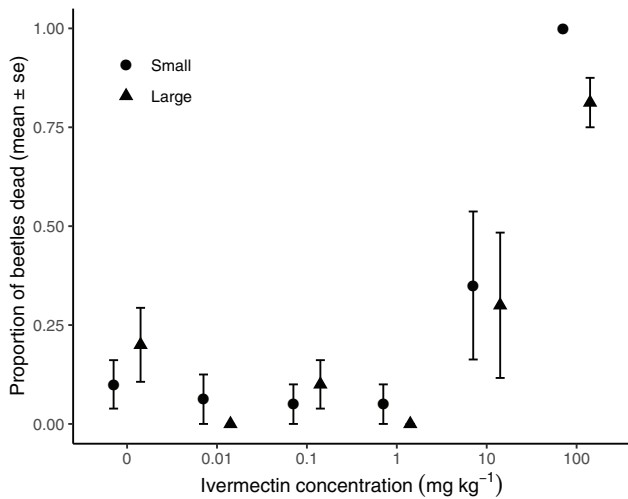

**Figure 3 High doses of ivermectin cause mortality in the dung beetle *Onthophagus nuchicornis*.** Mortality of *Onthophagus nuchicornis* exposed to ivermectin in dung. Mortality varied with ivermectin concentration, but body size did not affect susceptibility to ivermectin by itself, or via any exposure-body size interaction. Points are mean values with associated standard errors.

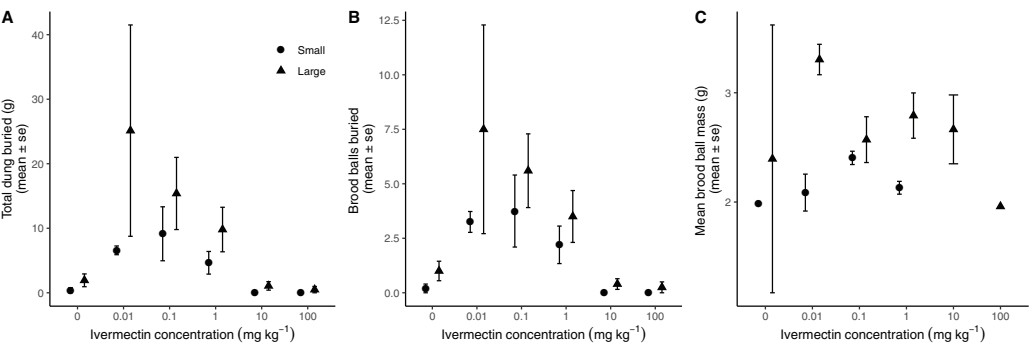

**Figure 4 Exposure to low doses of ivermectin stimulate functioning in the dung beetle *Onthophagus nuchicornis*.** Low exposure to ivermectin enhances the ecosystem function of dung burial. Larger beetles continue to be more functionally efficient than smaller beetles, regardless of ivermectin exposure. In each case, the higher level of functioning observed (A) was driven by greater brood ball production (B), rather than the production of heavier brood balls (C). Points represent means and associated standard errors.

The greater level of functional efficiency in large beetles can explained by two complementary factors. Larger beetles produced nearly twice as many brood balls as smaller beetles, and the mean size of brood balls by large beetles was 38% greater than brood balls produced by smaller beetles. These observed differences can be explained through biological and behavioral mechanisms.

The greater number of brood balls constructed by larger beetles may be directly related to fecundity. Female body-size is positively and strongly linked to fecundity across many insect groups (*Honěk, 1993*). However, in *O. nuchicornis*, both males and females

contribute to formation of brood balls (*Macqueen & Beirne, 1975a*), meaning the number of brood balls constructed by a pair of beetles reflects more than fecundity alone.

The larger size of brood balls can be explained by both behavioral and morphological differences amongst size classes. A behavioral explanation of these differences is linked to reproductive investment by females. A study of a closely-related species, *O. taurus*, found that female beetles formed larger brood balls, when mated to larger bodied males (*Hunt & Simmons, 2001*). Because we modified male and female size simultaneously, we are unable to test whether this effect occurred within our study species. Onthophagine dung beetles use their legs and bodies to combine small quantities of dung into tightly-packed masses known as brood balls. Because of their greater size, larger beetles should build larger brood balls in comparison to smaller beetles. Past research focusing on *O. gazella* found similar positive relationships between body size and brood ball size (*Lee & Peng, 1981*).

Although mortality of *O. nuchicornis* from ivermectin exposure was significant at concentrations of 10 mg·kg$^{-1}$ (~35% mortality) and 100 mg·kg$^{-1}$ (~90% mortality), larger body size continued to support higher levels of functional efficiency. That is, irrespective of ivermectin exposure, large beetles buried more dung than smaller beetles. Unlike in the first part of the experiment, this was caused solely by brood ball abundance and not by a combination of brood ball abundance and size. Comparing brood ball production from the first experiment with the control group in the second experiment, shows considerable loss of functional efficiency over time. This could indicate that laboratory conditions were not conducive to beetle health, or that field captured beetles were collected when their reproductive potential was declining.

Dung beetles exposed to low levels of ivermectin had greater functional efficiency and higher survival rates than beetles in the control treatment. This response is indicative of hormesis: a biphasic dose–response relationship where low doses of a stressor stimulate biological effects and high doses of the same stressor inhibit biological effects (*Southam, 1943*). While the phenomenon of hormesis has been observed across many different insect taxa (*Cohen, 2006*; *Cutler, 2013*; *Guedes & Cutler, 2014*), to the best of our knowledge this is the first time it has been recognized as such in dung beetles. Previous studies seem to have observed stimulatory effects of ivermectin on other dung beetle species, but the authors have not acknowledged the hormetic effects they reported. In a pair of studies by *Ishikawa & Iwasa (2020)*, exposure to 0.1 ppm ivermectin in cow dung increased brood ball production 32% in *Onthophagus bivertex* and 52% in *Onthophagus lenzii*. A similar finding was reported by *Iwasa et al. (2007)* who reported that exposure to 0.05–0.5 ppm (wet weight) ivermectin in cow dung increased brood ball formation by as much as 198% in the dung beetle *Caccobius jessoensis*, relative to an ivermectin-free control.

The range of ivermectin concentrations that stimulated a hormetic response in our experiment (0.01–1 mg kg$^{-1}$), are commonly encountered in agroecosystems. A pharmokinetic study of ivermectin in sheep (*Vokřál et al., 2019*), where animals were dosed with a subcutaneous injection of 0.2 mg ivermectin kg$^{-1}$ bodyweight, resulted in a maximum concentration of $0.93 \pm 0.50$ mg kg$^{-1}$ ivermectin in dung (wet weight) 48 h

following injection, which quickly declined to approximately 0.05 mg kg$^{-1}$ 12 days post treatment. Experiments with cattle also demonstrate the hormetic concentrations in our experiments also occur in the field. For example, after treating cattle with a pour-on application of 500 µg ivermectin kg bodyweight$^{-1}$, concentrations of ivermectin in dung were 0.97–1.01, 0.50–1.54 and 0.07–0.14 mg kg$^{-1}$ wet weight at 3, 7 and 14 days respectively (*Wohde et al., 2016*). Concentrations of ivermectin in dung can be above these sublethal concentrations depending on the product formulation (*Sommer et al., 1992*), species receiving the treatment (*Canga et al., 2009*), and occurrence of social grooming behaviors (*Laffont et al., 2001*).

We found that ecosystem functioning by a common and highly abundant species can be stimulated by exposure to low concentrations of ivermectin in dung. This has potentially far-reaching beneficial effects for pasture environments. By increasing the quantity of dung buried in soil, tunneling dung beetles like *O. nuchicornis* move greater quantities of nutrients into the plant root zone, which stimulates primary productivity. Increased burial also results in clearing of dung from the pasture surface, which frees additional pasture area for grazing livestock (*Anderson, Merritt & Loomis, 1984*). Dung burial by tunneling beetles is well known to prevent the development of blood-feeding veterinary pests such as *Haematobia irritans* (Diptera: Muscidae) (*Legner & Warkentin, 1991*). Furthermore, enhanced dung burial and physical disruption of the dung environment could work alongside the toxic effects of ivermectin on horn fly development, and might complement programs developed to manage resistance to ivermectin and other macrocyclic lactones (*Byford et al., 1999*).

While low-dose exposure to ivermectin stimulated higher levels of functioning than the controls, it is important to reiterate the loss of functional efficiency between the first and second experiment. Beetles that experienced no ivermectin-exposure in the second experiment buried on average just 2.2 ± 0.3 brood balls, whereas the same cohort of the beetles in the first experiment buried 10.6 ± 2.6 brood balls, representing a ~80% reduction between the two experiments. It seems likely that the experimental conditions were not conducive to prolonged insect health. The hormetic response of dung beetles to ivermectin during the second part of the experiment might not have been as pronounced in a less stressful environment.

Trade-offs may have occurred with biological endpoints not measured in our study. For example, others have shown that exposure of dung beetles to ivermectin in dung stimulates broodball formation, but thereafter results in reduced emergence of F1 offspring (*Ishikawa & Iwasa, 2020*). This may be because of greater ivermectin sensitivity in developing larvae relative to adult beetles, which has been reported in multiple studies (*Beynon et al., 2012*; *Pérez-Cogollo et al., 2015*). More work is needed to clarify whether ecological benefits of enhanced dung processing stemming from exposure to low doses of ivermectin results in increased functioning of the system overall, or neutralized or reduced functioning as a result of trade-offs.

Our findings must be considered against landscape-scale studies that indicate negative impacts of veterinary parasiticides on dung beetles. *Sands & Wall (2018)* sampled dung

beetles across 24 beef farms across Southwest England, finding farms that used macrocyclic lactones had 19% fewer species of dung beetles, but no reduced abundance, compared to farms that used no parasiticides. While not specifically targeting the effect of veterinary parasiticides, a study of 24 cattle farms in Ireland by *Hutton & Giller (2003)* found that organic farms supported dung beetle abundance 50% greater than conventional farms. Short-term benefits for ecosystem functioning caused by hormesis may be short-lived if dung beetle populations experience decline in abundance and diversity, both of which have been shown to negatively affect ecosystem functions provided by dung beetles (*Beynon et al., 2012*; *Manning & Cutler, 2018*).

Relatively little is known about how short-term exposure to parasiticide residues impacts beetles later in life. When *Aphodius fossor* fed on dung containing 0.5 mg kg$^{-1}$ ivermectin, dung removal was suppressed 48% relative to beetles feeding on ivermectin-free dung, but such effects subsided two weeks after exposure (*Manning, Beynon & Lewis, 2017*). If low-dose exposure to ivermectin can stimulate reproduction, hormetic effects could have a net positive effect on ecosystem functioning. Past research indicates this is possible: males of the dung beetle *Euoniticellus intermedius* exposed to 0.01 mg kg$^{-1}$ ivermectin in cattle dung had 39% increased testis volumes, which is indicative of higher fertility (*Hunt & Simmons, 2001*) and females exposed to the same ivermectin treatment had 25% more mature oocytes (a measure of fecundity), with no effect on the number of brood masses formed; a second measure of fecundity.

Our current understanding of insecticide-induced hormesis is mostly limited to laboratory conditions, despite the ramifications this phenomenon could have for ecosystems (*Cutler, 2013*). Dung insect communities perturbed by parasiticide residues could be an excellent model system to better understand the strength and significance of hormetic relationships in complex environments subject to ecological flux and chemical disturbance. In conjunction with feeding on dung of livestock, dung beetles support a wide range of ecosystem functions within agroecosystems that vary from enhancing primary productivity (*Manning et al., 2017*) to disrupting parasite development (*Sands & Wall, 2017*). This offers a range of endpoints and experimental designs that can be used to better understand hormetic influences of stressors on ecosystem functioning.

## CONCLUSIONS

We found that body size was a strong predictor of functional efficiency for the dung beetle *Onthophagus nuchicornis*. This benefit of larger body size for functional efficiency persisted, even when beetles were exposed to the veterinary anthelmintic ivermectin. High concentrations of ivermectin in dung (10–100 mg·kg$^{-1}$) caused mortality, but low concentrations (0.01–1 mg·kg$^{-1}$) stimulated the amount of dung buried via increasing the number of brood ball production. This hormetic response of Onthophagus dung beetles to ivermectin has been reported in earlier publications but was not recognized as hormesis. Because dung beetles support healthy agroecosystems by providing a number of ecological functions, we suggest that dung beetles could be a useful system for understanding the significance of hormesis within ecosystems.

## ACKNOWLEDGEMENTS

We wish to thank Alexa Stack-Mills for assistance in the field and Emma Gray and Janessa Rathgeber for assistance in the lab.

### Funding

This work was supported by a Killam Postdoctoral Fellowship and a Natural Sciences and Engineering Research Council of Council (NSERC) Postdoctoral Fellowship to Paul Manning and a NSERC Discovery Grant (RGPIN-2015-04639) to G. Christopher Cutler. The funders had no role in study design, data collection and analysis, decision to publish, or preparation of the manuscript.

### Grant Disclosures

The following grant information was disclosed by the authors:
Killam Postdoctoral Fellowship and a Natural Sciences and Engineering Research Council of Council (NSERC) Postdoctoral Fellowship to Paul Manning.
NSERC Discovery: RGPIN-2015-04639.

### Competing Interests

G. Christopher Cutler is an Academic Editor for PeerJ.

### Author Contributions

- Paul Manning conceived and designed the experiments, performed the experiments, analyzed the data, prepared figures and/or tables, authored or reviewed drafts of the paper, and approved the final draft.
- G Christopher Cutler conceived and designed the experiments, authored or reviewed drafts of the paper, and approved the final draft.

### Field Study Permissions

The following information was supplied relating to field study approvals (i.e., approving body and any reference numbers):

Ann Allen (Camden Stables) and Suzanne Perry (Opportunity Farm) permitted us to collect dung beetles on their property. Dalhousie Agricultural Campus Farm staff permitted us access to the farm to collect sheep dung.

### Data Availability

The data is available on Dataverse: Manning, Paul, 2020, "Dung_Beetle_Hormesis", https://doi.org/10.5683/SP2/AMMGSY, Scholars Portal Dataverse, V1, UNF:6: Q1L36fI7LB+LeF1KNBSEWg== [fileUNF]

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
