# Peer review of "Exposure to low concentrations of pesticide stimulates ecological functioning in the dung beetle Onthophagus nuchicornis"

_PeerJ, doi:10.7717/peerj.10359_

## Round 0.1 · original submission · Minor Revisions

Three reviewers as experts in their field have reviewed your manuscript and proposed a number of changes which you should follow in your revision. I look forward to receiving your revised manuscript.

Reviewer 1 ·

Basic reporting

42 Remove W.
72-75 Could benefit from a little more detail and more recent references e.g. pasture fouling, greenhouse gas reduction, and pest and parasite control. Although I cannot find studies which specifically focus on O. nuchicornis, similar benefits will be provided by other Onthophagus species for which studies have been completed.
96 Agreed
98-102 For predictions b and c please add reasoning behind these predictions.
170 paper is not in sections, remove.
170 I think this is clearer “The same beetles were used for the second experiment, which began immediately after the first experiment. Dung was defrosted forty-eight hours prior and subsequently homogenized for 10 min, as described above.”
182-183 Remove sentence as described above.
260 missing full stop.
339 beetle*s*

Experimental design

116 was this horse dung collected from the same facilities? Please specify.
124 how was the dung homogenised?
136 where was the builders sand from, could this have contained any pollutants? Possible cause of high mortality in the control. Worth using children’s play sand in subsequent experiments.
I’m assuming there was no mortality during the first experiment? State this here.
178 – was the paint mixed cleaned before each homogenisation to remove residue ivermectin? How was it cleaned? Was the control group homogenised first?

Validity of the findings

For all analyses, why are beetles divided into ‘small’ and ‘large’? Please give reasoning for not using size (pronotal width) as a continuous trait.
268-269 would be useful to state how ivermectin concentration affected survival in text.
274 Surprising result! Add the effect at the two highest concentrations of ivermectin.
363 What kind of treatment was this? I think it is important to mention that higher concentration of IVM will be present in dung if the treatment is applied topically eg Sommer et al. 1992. Here they found concentrations of 9 mg kg-1 dry weight 24 hours after treatment. This translates to around 2 mg kg-1 wet weight if dung is 20% dry matter.
Sommer, C., Steffansen, B., Nielsen, B., Grønvold, J., Jensen, V.K., Jespersen, B.J., Springborg, J. and Nansen, P. (1992). Ivermectin excreted in cattle dung after subcutaneous injection or pour-on treatment: Concentrations and impact on dung fauna. Bulletin of Entomological Research, 82, 257-264.
365-377 Good to mention here high mortality in control group.
365-377 Although long-term studies looking at several generations of dung beetles have not been completed (to my knowledge), field studies have which compare organic and conventional farms. For example:
Sands B and Wall R (2018) Sustained parasiticide use in cattle farming affects functionally important dung beetle assemblages. Agriculture Ecosystems and Environment 265, 226–235.
I think the possible long-term impact of ivermectin on dung beetles should be mentioned with more emphasis.

Additional comments

Overall an enjoyable read with interesting results. Clear aims and results.

Reviewer 2 ·

Basic reporting

Line 162 remove space between for and seven

Line 176 replace or with and

Line 303 space before "The greater..."

Line 401 please correct "suppressed"

Please check reference list, e.g. Wohde et al (2016) is cited in line 364 but not listed in reference chapter.

Experimental design

Add moisture content of sheep dung at least or in best case add concentrations of ivermectin also per dry weight dung. This might increase comparability with other studies.

Validity of the findings

No comments

Additional comments

No comments

·

Basic reporting

no comment

Experimental design

No comment

Validity of the findings

No comment

Additional comments

This is a well written manuscript and I can find very little to criticize. The experimental approach is nice although, as the authors acknowledge, there are some areas in which it could have been made larger and more comparisons could have been included. The authors do a very nice job of bring in past literature and showing how it could have been used in a similar set of comparisons. This manuscript is also makes use of the data in an ecological manner which is a rather novel approach to using data which is more often used as toxicological information.

I have a number of minor queries that should be addressed.

The following is from the PeerJ guide and should be followed throughout. “For three or fewer authors, list all author names (e.g. Smith, Jones & Johnson, 2004). For four or more, abbreviate with ‘first author’ et al. (e.g. Smith et al., 2005).”

Does dung beetle burial of dung ‘impair development’ of horn fly larvae? I find that this statement is rather odd. I suspect the authors are a bit to cautious; why not call a spade a spade.

Use of the phrase ‘due to’ is not correct. This phrase is used in banking to refer to funds that are ‘due to’ an account and is not indented to be used in any other context.

---

## Round 0.2 · accepted · Accept

Thank you very much for the thorough revision.

Reviewer 1 ·

Basic reporting

N/A

Experimental design

N/A

Validity of the findings

N/A

Additional comments

Thank you to the authors for addressing each of my comments. I am now satisfied that the paper is of publishable quality.

---

## Author Rebuttal · Round 0.2

# DALHOUSIE UNIVERSITY

October 6th, 2020

Dear Professor Oehlmann,

I am writing to re-submit our revised article "Exposure to low concentrations of pesticide stimulates ecological functioning in the dung beetle *Onthophagus nuchicornis*" to PeerJ. My co-author and I are grateful to the reviewers for their very constructive comments and suggestions.

On the next page, we have incorporated the feedback from each of the peer-reviewers and have provided line references that indicate where in the manuscript these changes were made. These line references correspond to the "marked up" version of the manuscript. The reference section has been completely overhauled to better reflect the style guide of PeerJ. This is reflected in the Clean version of the document.

Thank-you for considering our paper for publication, and please do not hesitate to reach out if any clarifications are required.

Sincerely,

Paul Manning

**Reviewer #1**

42 Remove W.
**DONE [L42]**

72-75 Could benefit from a little more detail and more recent references e.g. pasture fouling, greenhouse gas reduction, and pest and parasite control. Although I cannot find studies which specifically focus on O. nuchicornis, similar benefits will be provided by other Onthophagus species for which studies have been completed.
**DONE. Have included more ecosystem functions supported by Onthophagine beetles. Have expanded beyond O. nuchicornis [L74].**

98-102 For predictions b and c please add reasoning behind these predictions.
**DONE [L101-112] We have contextualized the second and third prediction.**

170 paper is not in sections, remove.
**DONE [L187]**

170 I think this is clearer "The same beetles were used for the second experiment, which began immediately after the first experiment. Dung was defrosted forty-eight hours prior and subsequently homogenized for 10 min, as described above."
**DONE. Paragraph beginning on L183 has been rewritten based on your suggestion.**

182-183 Remove sentence as described above.
**DONE. Addressed when rewriting the paragraph beginning on L183.**

260 missing full stop.
**DONE [L282]**

339 beetle*s*
**DONE [L366]**

116 was this horse dung collected from the same facilities? Please specify.

**DONE. Yes it was from the same facilities, this is specified on L125.**

124 how was the dung homogenised?

**INFORMATION ADDED. Using a paint mixer attachment for c. 10 minutes [L134]**

136 where was the builders sand from, could this have contained any pollutants? Possible cause of high mortality in the control. Worth using children's play sand in subsequent experiments.

**INFORMATION ADDED. Sand was from a local quarry. Children's play sand is a useful suggestion that we will consider for future experiments. [L146]**

I'm assuming there was no mortality during the first experiment? State this here.

**INFORMATION ADDED There were three mortalities in the first part of the experiment that were replaced with beetles held under identical conditions on "standby", this has been added in [L180]**

178 – was the paint mixed cleaned before each homogenisation to remove residue ivermectin? How was it cleaned? Was the control group homogenised first?

**DONE. Information on homogenization and cleaning of equipment was added [L201]**

For all analyses, why are beetles divided into 'small' and 'large'? Please give reasoning for not using size (pronotal width) as a continuous trait.

**RESPONSE. This is a good suggestion for an alternative analysis. Our categorical approach was validated by our comparison of body-size categories (pronotal width and body mass), and is straight-forward to understand for the reader. Our approach is specifically why we considered beetles from the extremes of the sample population.**

268-269 would be useful to state how ivermectin concentration affected survival in text

**DONE. This has been added in on L292.**

274 Surprising result! Add the effect at the two highest concentrations of ivermectin.

**DONE [L304]. Please note that we caught an error when reporting control treatments values, which affects the magnitude of the perturbation on functioning.**

363 What kind of treatment was this? I think it is important to mention that higher concentration of IVM will be present in dung if the treatment is applied topically eg Sommer et al. 1992. Here they found concentrations of 9 mg kg-1 dry weight 24 hours after treatment. This translates to around 2 mg kg-1 wet weight if dung is 20% dry matter.

**Response: DONE L392. Wohde et al reports a pour-on treatment. Context about higher concentrations, and their causes has been added, along with the reference to Sommer et al 1992.**

365-377 Good to mention here high mortality in control group.

**DONE. Added in greater detail [L292] in the results about the control mortality (15%)**

365-377 Although long-term studies looking at several generations of dung beetles have not been completed (to my knowledge), field studies have which compare organic and conventional farms. For example:

Sands B and Wall R (2018) Sustained parasiticide use in cattle farming affects functionally important dung beetle assemblages. Agriculture Ecosystems and Environment 265, 226–235.

I think the possible long-term impact of ivermectin on dung beetles should be mentioned with more emphasis.

**DONE. This is a useful suggestion, we have added a paragraph adding this context to the discussion [L435]**

Overall an enjoyable read with interesting results. Clear aims and results.

**Response: We appreciate your constructive comments. Thank-you.**
* * *
**Reviewer #2**

Basic reporting

Line 162 remove space between for and seven

**DONE [L173]**

Line 176 replace or with and

**DONE [L194]**

Line 303 space before "The greater..."

**DONE [329]**

Line 401 please correct "suppressed"

**DONE [L450]**

Please check reference list, e.g. Wohde et al (2016) is cited in line 364 but not listed in reference chapter.

**DONE. Wohde et al (2016) been added to the reference list.**

Experimental design

Add moisture content of sheep dung at least or in best case add concentrations of ivermectin also per dry weight dung. This might increase comparability with other studies.

**Unfortunately, we did not measure the moisture content in dung. All measures remain reported as mg ivermectin · kg$^{-1}$ dung (wet weight).**

Validity of the findings

No comments

Comments for the Author

No comments
* * *
**Reviewer #3 (Doug Colwell)**

Basic reporting

no comment

Experimental design

No comment

Validity of the findings

Comments for the Author

This is a well written manuscript and I can find very little to criticize. The experimental approach is nice although, as the authors acknowledge, there are some areas in which it could have been made larger and more comparisons could have been included. The authors do a very nice job of bring in past literature and showing how it could have been used in a similar set of comparisons. This manuscript is also makes use of the data in an ecological manner which is a rather novel approach to using data which is more often used as toxicological information.

I have a number of minor queries that should be addressed.

The following is from the PeerJ guide and should be followed throughout. "For three or fewer authors, list all author names (e.g. Smith, Jones & Johnson, 2004). For four or more, abbreviate with 'first author' et al. (e.g. Smith et al., 2005)."
**DONE. Amended as suggested for all references. The entire reference section has been carefully checked to ensure it matches the PeerJ specifications. This has been amended within the cleaned version of the manuscript.**

Does dung beetle burial of dung 'impair development' of horn fly larvae? I find that this statement is rather odd. I suspect the authors are a bit to cautious; why not call a spade a spade.
**DONE. [L408] The sentence has been updated to better reflect this ecological relationship.**

Use of the phrase 'due to' is not correct. This phrase is used in banking to refer to funds that are 'due to' an account and is not indented to be used in any other context.
Response: **Amended throughout as suggested. [L243, L349, L428]**